# The Efficiency of Industrial and Laboratory Anaerobic Digesters of Organic Substrates: The Use of the Biochemical Methane Potential Correction Coefficient

**Krzysztof Pilarski [1], Agnieszka A. Pilarska [2],\*, Piotr Boniecki [1], Gniewko Niedbała [1], Karol Durczak [1], Kamil Witaszek [1], Natalia Mioduszewska [1] and Ireneusz Kowalik [1]**

[1]  Institute of Biosystems Engineering, Poznań University of Life Sciences, Wojska Polskiego 50, 60-637 Poznań, Poland; pilarski@up.poznan.pl (K.P.); bonie@up.poznan.pl (P.B.); gniewko.niedbala@up.poznan.pl (G.N.); kdurczak@up.poznan.pl (K.D.); kamil.witaszek@up.poznan.pl (K.W.); natalia.mioduszewska@up.poznan.pl (N.M.); ikowalik@up.poznan.pl (I.K.)

[2]  Institute of Food Technology of Plant Origin, Poznań University of Life Sciences, Wojska Polskiego 31, 60-637 Poznań, Poland

\*  Correspondence: pilarska@up.poznan.pl; Tel.: +48-61-848-73-08

**Abstract:** This study is an elaboration on the conference article written by the same authors, which presented the results of laboratory tests on the biogas efficiency of the following substrates: maize silage (MS), pig manure (PM), potato waste (PW), and sugar beet pulp (SB). This article presents methane yields from the same substrates, but also on a technical scale. Apart from that, it presents an original methodology of defining the Biochemical Methane Potential Correction Coefficient (BMPCC) based on the calculation of biomass conversion on an industrial scale and on a laboratory scale. The BMPCC was introduced as a tool to enable uncomplicated verification of the operation of a biogas plant to increase its efficiency and prevent undesirable losses. The estimated BMPCC values showed that the volume of methane produced in the laboratory was overestimated in comparison to the amount of methane obtained under technical conditions. There were differences observed for each substrate. They ranged from 4.7% to 17.19% for MS, from 1.14% to 23.58% for PM, from 9.5% to 13.69% for PW, and from 9.06% to 14.31% for SB. The BMPCC enables estimation of biomass under fermentation on an industrial scale, as compared with laboratory conditions.

**Keywords:** laboratory-scale efficiency; industrial-scale efficiency; biomass conversion; Biochemical Methane Potential Correction Coefficient; loss prevention

## 1. Introduction

The intensive development of agriculture causes an increase in the supply of organic waste. Waste matter disposal technologies are often based on the anaerobic digestion (AD) process, which takes place in biogas plants. In order to increase the efficiency of the installation, plants with high content of organic matter are added as a substrate [1,2]. Target crops are grown for this purpose—mainly maize and grass, which are used to produce silages [3]. Animal waste [4,5] and food waste [6,7] are also used as substrates. Biogas plants produce biogas, which is classified as a source of renewable energy because it contains methane. The resulting gas, which is an energy carrier, can be easily converted into electricity and heat.

The AD process is one of the most adequate and prospective methods of organic waste disposal and it is also a source of biofuel [8,9]. However, the process must be economically viable and stable. The main factors which affect AD-process efficiency are as follows: the chemical composition of the substrates, pH, temperature, the substrates mixing process, dry residue, the content of organic matter

in the substrate, the digester load, and the concentration of inhibitors [10–12]. These parameters are controlled both in the laboratory-scale and industrial-scale process.

A slight change in the conditions of the methane digestion in a biogas plant may upset the process or halt it completely. Temperature is one of the factors that significantly affects the course of the digestion process. Small changes in temperature and the retention time affect bacterial activity [13]. As a consequence, biogas production is reduced. In Poland, the digestion process is usually carried out under mesophilic conditions at a temperature of about 39 °C. In Europe, mesophilic installations are predominant, but there are also thermophilic conditions, where more heat energy is used by the plant to sustain the process [14,15]. As mentioned before, pH is also important for methane fermentation. It should range from 6.8 to 7.5 because this guarantees optimal conditions for bacterial life and reproduction during all four phases of the methane fermentation process [16,17]. The biogas yield decreases when the pH value is higher than 7.5 or lower than 6.8.

When biomass is applied to digestion chambers with high-protein substrates, excessive amounts of ammoniacal nitrogen are usually formed. Excess ammonia significantly slows down the biogas production process. As temperature increases, so does the effect of ammonia, inhibiting methane fermentation [18,19]. Some substrates are rich in sulphur and they cause excessive amounts of this element in the digester. The element may occur in the form of ions dissolved during the liquid phase, or it may be found in hydrogen sulphide in a mixture of liquid and gas. Higher temperature increases the solubility of hydrogen sulphide, which results in its higher concentration during the liquid phase [20]. At the initial stage of operation of a biogas plant, it is very important to supply small portions of substrates with a homogeneous composition. This affects the overall normal digestion by individual bacteria at each stage of the fermentation process [21]. During the operation of a biogas plant, it is necessary to observe the retention time for individual organic substrates. This is the time a given substrate should spend in the digester until it achieves an appropriate level of degradation. These are the most common problems encountered in practice, which directly affect the actual production of biogas.

The pursuit of more efficient use of biomass to generate energy requires detailed verification of methane fermentation technologies. Detailed analysis and the selection of appropriate biogas production technologies gives investors a real opportunity to manage biogas plants effectively. Apart from that, more efficient and effective installations give a chance to reduce the financial and social costs of every kilowatt hour (kWh) of energy produced, as compared with the countries regarded as pioneers in this field. Poland is still a developing country, as it is attempting to catch up with the standard of Western European countries. Therefore, it should effectively control the expending of funds on solutions implemented in the field of renewable energy sources [1,12,22].

At the beginning of the development of biogas installations in Poland, i.e., between 2008 and 2009, investors were ready to expend money on large plants capable of generating a power of 2 MW or more [23]. However, the market very quickly verified these plans and showed that it was the wrong trend because it involved high costs of transport (it was necessary to supply thousands of tonnes of substrates). The dispersion of Polish agriculture was not taken into consideration either. After about 3 years, investors began to prefer biogas plants capable of generating a power of 0.5–1 MW. They were usually located on large farms [24] because it was necessary to provide a continuous supply of substrates and to solve the deodorisation problem. However, this method was also unsuccessful because the actual fermentation efficiency was much lower than the forecast. The energy consumption for fittings was higher than assumed, mostly due to the increased and unstable operation of the mixing systems in the installation [23]. These problems are still topical. Therefore, the proposed technologies should be analysed in detail and adjusted to national and local requirements. Scientists and biogas plant owners struggle with these problems not only in Poland but also in other countries [25–28].

So far, researchers have not prepared the assumptions that would clearly and easily verify the operation of a biogas plant. This verification system is more and more wanted by current and future investors. As seen in reference publications, most studies conducted by scientific institutions both

in Poland and other countries usually refer to laboratory conditions [29–31]. The authors of most of these studies did not analyse the operation of biogas installations, which converted biomass into biogas/methane. However, these results directly translate into the economic and ecological effect of bioelectric plants. In general, scientific reports very rarely provide information about the efficiency of a particular technology proposed by researchers in terms of the relation between the amount of biogas generated in a laboratory and the potential efficiency of this technology applied on a technical scale. If any information is given, it is usually insufficient.

This study compares the efficiency of the anaerobic digestion technology applied on an industrial scale with its efficiency on a laboratory scale. The comparison was made for the same substrates. The study also verifies the level of significance of the determinants affecting the efficiency of operation of the biogas installation. The following substrates were used in the research: maize silage (MS); and agri-food waste, including pig manure (PM), potato waste (PW), and sugar beet pulp (SB). In view of the fact that numerous dependencies can be observed during the AD process in a biogas plant—and due to the fact that reference publications do not assess the efficiency of biomass conversion on an industrial scale—the Biochemical Methane Potential Correction Coefficient (BMPCC) was defined [23]. The coefficient provided information about the efficiency of organic matter decomposition into methane in a biogas plant and enabled comparison of these results with the AD process conducted in a laboratory.

## 2. Materials and Methods

### 2.1. Materials

Green substrates—maize silage (MS) and agri-food waste, pig manure (PM), potato waste (PW), and sugar beet pulp (SB) were used in the study. The waste materials were acquired from a farm and a sugar factory in the Wielkopolska region, Poland.

Maize silage was stored in silos, where it was ensilaged. Pig manure was supplied directly from a pig farm. Potato waste was also supplied directly after being customised. Sugar beet pulp was stored in silage sleeves. The degree of maize compaction in the ensilage process, the weather conditions, and storage time were the factors affecting the physicochemical properties after storage.

### 2.2. Physicochemical Analysis of Materials

The following standards were used in physicochemical analyses of the substrates (used organic materials) and samples (harvested fermentation mixture): pH—the potentiometric method, PN-EN 12176: 2004; dry residue—the weight method, PN-EN 12880: 2004; roasting losses (roasting residue) —the weight method, PN-EN 12879: 2004; sampling for chemical and physical tests, PN-EN ISO 5667-13: 2011; carbon, EN ISO 16948: 2015; hydrogen, EN ISO 16948: 2015; nitrogen, EN ISO 16948: 2015; oxygen, based on calculations; sulphur, PN-EN ISO 11885: 2009.

### 2.3. Laboratory-Scale Biogas Production

The anaerobic digestion process was conducted in a periodic mode of operation of digesters, under mesophilic conditions. The authors of this study presented a detailed diagram and described the construction and operation of biodigesters in their previous publications [32–34].

### 2.4. The Construction and Operation of an Industrial Installation

The biomass conversion tests were conducted for 6 months in an agricultural biogas plant. The facility was fed with the substrates listed above. Samples of the substrates were collected at monthly intervals because their physicochemical properties may have changed during storage.

The biogas plant consisted of two main digesters (F1 and F2) and a third tank, where digestate pulp was stored. The first tank (F1) was used for primary digestion, and the second tank (F2) was used for secondary digestion. The installation was also equipped with a primary tank (PT), into which waste potatoes were fed because they may have been contaminated with soil. Pig manure, which was

the agent liquefying the entire fermenting mass, was also fed into this tank. The annual production capacity was about 3.5 million m$^3$ of biogas. The installation was equipped with a cogeneration system, whose power was 1 MW$_{el}$.

### 2.5. Collection of Samples for Tests

Samples for tests were collected once a month. Six samples of each substrate were tested during the six-month experiment. The following aspects were taken into account when collecting the samples: access to the sampling point, the possibility to safely interrupt the stream of material when samples were collected manually, and the type of construction of the fermentation chamber—due to the stratification of the material collected for tests. The safest and most practical station for manual sample collection was selected. The practicality of this location was analysed in terms of the representativeness of the material collected for tests. Each time, at least 3 samples were collected to increase confidence in the representativeness of the material collected.

### 2.6. Qualitative and Quantitative Analysis of Biogas

The quality and quantity of the biogas produced was analysed once a day. In order to make effective measurements of the biogas quality, a minimum quantity of 0.4 L had to be produced daily. When a smaller amount of biogas was produced, the chemical composition was not analysed. The measurement methodology was adopted from the German standard DIN 38414/S8, which was modified by the author in order to reduce measurement errors [35]. The qualitative composition of biogas was analysed by measuring the content of $CH_4$, $CO_2$, $NH_3$, and $H_2S$.

The quality and quantity of the biogas produced in the installation under real conditions was based on the measuring systems in place. These systems were permanently installed and met the German standard DIN 38414/S8. There were no breakdowns during the entire period under study. When estimating the uncertainty of measurement in this article, the procedures followed Polish standards and German standard [35–37].

### 2.7. Biochemical Methane Potential Correction Coefficient (BMPCC)—Calculation Methodology Based on

First, the yield of biogas from the substrate was measured (m$^3$·Mg$^{-1}$ fresh matter (FM)) in a laboratory. Simultaneously, the composition of biogas was analysed by measuring the content of $CH_4$ and $CO_2$ (the concentrations of $NH_3$ and $H_2S$ were omitted). Then, the biogas composition was used to calculate the volume of methane. At the next stage, the mass of methane contained in the biogas was measured under laboratory conditions (the mass of methane in the biogas obtained from the fresh matter of the substrate under laboratory conditions, MMB-L). The third stage involved analysing the substrate for dry residue, roasting losses, and the content of carbon, hydrogen, oxygen, nitrogen, and sulphur. Then, knowing the content of C, H, O, N, and S, the amount of methane that could theoretically be obtained was calculated (theoretical methane mass, TMM) according to the principle of mass conservation.

The fourth stage involved calculation of the conversion of organic matter contained in the biomass under laboratory conditions (conversion of organic matter under laboratory conditions—the laboratory degree of biomass conversion, COM-L). The following Equation (1) was used:

$$COM - L = \frac{MMB - L}{TMM}. \tag{1}$$

The fifth stage involved calculation of the conversion of organic matter contained in the biomass under the operating conditions of the installation (conversion of organic matter in the installation—the industrial degree of biomass conversion, COM-I). The following Equation (2) was used:

$$COM - I = \frac{MMB - I}{TMM}. \tag{2}$$

At the last stage, the BMPCC of each substrate was calculated as the ratio between the mass of methane produced in the installation and the mass of methane produced under laboratory conditions. The following Equation (3) was used:

$$\text{BMPCC} = 100 - \frac{\text{COM} - \text{I}}{\text{COM} - \text{L}} \times 100. \tag{3}$$

## 3. Results

### 3.1. Physicochemical Parameters of Substrates—Laboratory-Scale Measurements

#### 3.1.1. pH of Substrates

The pH of maize silage used for the tests ranged from 4.21 (1MS) to 4.39 (3MS). The pH values of pig manure were similar during the entire test and ranged from 7.22 (5PM) to 7.56 (1PM). They were similar to data provided in reference publications [38,39]. The concentration of hydrogen ions in waste potatoes ranged from 7.44 (3PW) to 7.78 (4PW). As seen in results from the research on methane fermentation of waste potatoes conducted by [40], the pH of the substrate was 7. The pH of beet pulp ranged from 5.01 (1SB) to 5.18 (3SB). Table 1 shows the pH values of the substrates.

**Table 1.** Physicochemical properties and the laboratory-scale biogas efficiency of the substrates with the uncertainty of results, based on [1].

| Substrate | pH (-) | TS (%) | VS (%) | Biogas $(m^3 \cdot Mg^{-1}$ FM) | Biogas $(m^3 \cdot Mg^{-1}$ TS) | Biogas $(m^3 \cdot Mg^{-1}$ VS) | $CH_4$ (%) |
|---|---|---|---|---|---|---|---|
| **1MS** | 4.21 ± 0.06 | 32.68 ± 0.51 | 95.15 ± 1.76 | 188 ± 3.6 | 575 ± 12.6 | 605 ± 13.4 | 51.2 ± 1.29 |
| **2MS** | 4.28 ± 0.06 | 32.21 ± 0.50 | 94.61 ± 1.75 | 183 ± 3.5 | 568 ± 12.4 | 601 ± 13.4 | 52.3 ± 1.31 |
| **3MS** | 4.39 ± 0.06 | 32.11 ± 0.50 | 94.21 ± 1.75 | 181 ± 3.5 | 564 ± 12.3 | 598 ± 13.3 | 50.4 ± 1.27 |
| **4MS** | 4.31 ± 0.06 | 31.86 ± 0.50 | 93.65 ± 1.74 | 178 ± 3.4 | 559 ± 12.2 | 595 ± 13.2 | 50.9 ± 1.28 |
| **5MS** | 4.35 ± 0.06 | 31.45 ± 0.49 | 94.83 ± 1.76 | 180 ± 3.5 | 572 ± 12.5 | 604 ± 13.4 | 51.6 ± 1.30 |
| **6MS** | 4.28 ± 0.06 | 31.06 ± 0.48 | 93.88 ± 1.74 | 184 ± 3.5 | 592 ± 12.9 | 631 ± 14.0 | 50.4 ± 1.27 |
| **1PM** | 7.56 ± 0.10 | 4.86 ± 0.08 | 76.16 ± 1.41 | 17 ± 0.3 | 350 ± 7.6 | 459 ± 10.2 | 52.6 ± 1.32 |
| **2PM** | 7.44 ± 0.10 | 4.32 ± 0.07 | 76.88 ± 1.43 | 19 ± 0.4 | 440 ± 9.6 | 572 ± 12.7 | 51.1 ± 1.28 |
| **3PM** | 7.31 ± 0.10 | 4.94 ± 0.08 | 78.49 ± 1.46 | 20 ± 0.4 | 405 ± 8.9 | 516 ± 11.5 | 51.8 ± 1.30 |
| **4PM** | 7.28 ± 0.10 | 4.65 ± 0.07 | 81.32 ± 1.51 | 22 ± 0.4 | 473 ± 10.3 | 582 ± 12.9 | 51.4 ± 1.29 |
| **5PM** | 7.22 ± 0.10 | 5.06 ± 0.08 | 80.11 ± 1.49 | 21 ± 0.4 | 415 ± 9.1 | 518 ± 11.5 | 51.3 ± 1.29 |
| **6PM** | 7.36 ± 0.10 | 5.01 ± 0.08 | 79.84 ± 1.48 | 18 ± 0.3 | 359 ± 7.8 | 450 ± 10.0 | 50.8 ± 1.28 |
| **1PW** | 7.36 ± 0.10 | 21.31 ± 0.33 | 94.87 ± 1.76 | 68 ± 1.3 | 319 ± 7.0 | 336 ± 7.5 | 51.6 ± 1.30 |
| **2PW** | 7.41 ± 0.10 | 21.33 ± 0.33 | 94.61 ± 1.75 | 69 ± 1.3 | 323 ± 7.1 | 342 ± 7.6 | 50.7 ± 1.27 |
| **3PW** | 7.44 ± 0.10 | 21.45 ± 0.33 | 94.83 ± 1.76 | 67 ± 1.3 | 312 ± 6.8 | 329 ± 7.3 | 51.4 ± 1.29 |
| **4PW** | 7.78 ± 0.11 | 21.85 ± 0.34 | 95.01 ± 1.76 | 70 ± 1.3 | 320 ± 7.0 | 337 ± 7.5 | 51.1 ± 1.28 |
| **5PW** | 7.65 ± 0.10 | 21.78 ± 0.34 | 95.12 ± 1.76 | 71 ± 1.4 | 326 ± 7.1 | 343 ± 7.6 | 52.2 ± 1.31 |
| **6PW** | 7.71 ± 0.10 | 21.86 ± 0.34 | 95.92 ± 1.78 | 70 ± 1.3 | 320 ± 7.0 | 334 ± 7.4 | 51.8 ± 1.30 |
| **1SB** | 5.01 ± 0.07 | 23.88 ± 0.37 | 94.16 ± 1.75 | 99 ± 1.9 | 415 ± 9.1 | 440 ± 9.8 | 50.2 ± 1.26 |
| **2SB** | 5.08 ± 0.07 | 23.44 ± 0.36 | 94.02 ± 1.74 | 97 ± 1.9 | 414 ± 9.0 | 441 ± 9.8 | 50.8 ± 1.28 |
| **3SB** | 5.18 ± 0.07 | 23.58 ± 0.37 | 93.88 ± 1.74 | 96 ± 1.8 | 407 ± 8.9 | 434 ± 9.6 | 51.4 ± 1.29 |
| **4SB** | 5.16 ± 0.07 | 23.41 ± 0.36 | 93.32 ± 1.73 | 93 ± 1.8 | 397 ± 8.7 | 426 ± 9.5 | 52.1 ± 1.31 |
| **5SB** | 5.11 ± 0.07 | 23.67 ± 0.37 | 93.46 ± 1.73 | 94 ± 1.8 | 398 ± 8.7 | 425 ± 9.4 | 50.7 ± 1.27 |
| **6SB** | 5.09 ± 0.07 | 23.33 ± 0.36 | 94.12 ± 1.74 | 95 ± 1.8 | 407 ± 8.9 | 433 ± 9.6 | 51.5 ± 1.29 |

#### 3.1.2. Total Solids in Substrates

The content of total solids (TS) in the maize silage used in the installations ranged from 32.21% (2MS) to 31.06% (6MS). The TS content in the pig manure ranged from 4.32% (2PM) to 5.06 (6PM) and was consistent with the data provided in reference publications [41]. The TS content in the potato waste ranged from 21.33% (2PW) to 21.86% (6PW). The TS content in the beet pulp ranged from 23.33% (6SB) to 23.88% (1SB). The results were comparable with the data published in the study by [42]. Table 1 shows the TS content in the substrates.

### 3.1.3. Volatile Solids in Substrates

The content of volatile solids (VS), i.e., roasting losses, was another parameter under analysis. Substrates with high content of organic matter are a valuable raw material for biogas installations. There are three basic groups of organic matter in substrates: carbohydrates, protein, and fats. The materials used in this study mainly contained sugars and protein. The roasting losses for maize silage ranged from 93.88% (6MS) to 95.15% (1MS). The VS content in the pig manure ranged from 76.88% (2PM) to 81.32% (4PM). These values were similar to the data reported in reference publications [43,44]. The quality of liquid manure depends on the animals it comes from, their diet, and the degree of dilution with water. The VS content in the potato waste ranged from 94.61% (1PW) to 95.92% (6PW). Due to the content of carbohydrates in sugar beet pulp, this material positively influences the methane fermentation efficiency per digester volume unit. The VS content in the material used in the tests ranged from 93.32% (4SB) to 94.16% (1SB). Table 1 lists the results of this experiment.

### 3.2. Laboratory-Scale Biogas Efficiency of Samples

### 3.2.1. Volume of Biogas Obtained from Substrates per Fresh Matter

The yield of biogas obtained from maize silage amounted to 183 m³·Mg⁻¹ FM. The biogas volumes obtained in the experiment were consistent with the data provided by other researchers [33]. The yield of biogas obtained from liquid manure amounted to 20.1 m³·Mg⁻¹ FM. It was lower than the data provided in reference publications [45]. The volume of biogas obtained from waste potatoes amounted to 69.2 m³·Mg⁻¹ FM. The volume of biogas obtained from the fresh matter of sugar beet pulp amounted to 96.4 m³·Mg⁻¹ FM. There were similar values reported in reference publications [46,47]. Table 1 provides data on the volume of biogas obtained from the substrates.

### 3.2.2. Volume of Biogas Obtained from Substrates per Total Solids Content

Literature data provide the volume of biogas obtained in the AD process as the number of total solids contained in the samples so as to standardise the results without water. In this study, similar calculations were also performed. The volume of biogas obtained from maize silage at individual sample collection terms ranged from 559 (4MS) to 592 (6MS) m³·Mg⁻¹ TS. The yield of biogas obtained from pig manure per TS in individual samples ranged from 350 (3PM) to 473 (4PM) m³·Mg⁻¹ TS. In reference publications, there are big differences in the data on the biochemical methanogenic potential of pig manure obtained per TS [48]. The yield of biogas obtained from waste potatoes ranged from 323 (3PW) to 326 (5PW) m³·Mg⁻¹ TS. The biochemical methanogenic potential of sugar beet pulp ranged from 397 (4SB and 5SB) to 415 (1SB) m³·Mg⁻¹ TS. The volume of biogas obtained in this study was lower than in the study by [46]. The data on the volume of biogas per total solids content are shown in Table 1.

### 3.2.3. Volume of Biogas Obtained from Substrates per Volatile Solids Content

The yield of biogas obtained from maize silage per VS ranged from 595 (4MS) to 631 (6MS) m³·Mg⁻¹ VS. The results of this experiment were in line with the data presented by [44]. The yield of biogas obtained from pig manure ranged from 450 (6PM) to 582 (3PM) m³·Mg⁻¹ VS. The biogas volume was comparable to the results presented by [45]. The volume of biogas obtained from waste potatoes ranged from 329 (3PW) to 342 (2PW). The volume of biogas obtained from beet pulp ranged from 425 (5SB) to 440 (1SB and 2SB) m³·Mg⁻¹ VS. The values reported in reference publications were lower than the results presented by [46], where the biochemical methanogenic potential of sugar beet pulp amounted to 504 m³·Mg⁻¹ VS. The range of the results is given in Table 1.

The volume concentration of methane in the biogas obtained from the maize silage ranged from 50.4% (3MS) to 52.3% (2MS), see Table 1. The concentration noted in the presented experiment was identical with the data reported by [44]. The methane content in the biogas obtained from pig manure ranged from 50.8 (2PM) to 52.6% (5PM). According to the authors of scientific studies, the methane

fraction in the biogas obtained as a result of the methane fermentation of pig manure was above 53% [48]. The concentration of methane in the biogas obtained from waste potatoes ranged from 50.7% (2PW) to 52.2% (5PW). These values were similar to the data presented by [49]. The volume concentration of methane in the biogas obtained from beet pulp ranged from 50.2% (1SB) to 52.8% (4SB). According to [50], the concentration is usually about 52%.

The pH range corresponding to the maize silage proved that the ensilage process was successful. During storage, the total solids as well as volatile solids of this substrate were reduced. It is most likely that it was caused by microorganisms responsible for the conservation of MS. However, despite the decrease in the MS organic matter, the amount of biogas produced in the following months was not reduced. The physicochemical properties of pig manure depend on the way animals are fed and their age. This significantly affects the biomethane efficiency of this material. It is similar with potato waste, which is a waste material but is not cultivated for energy purposes. However, as the table shows, this did not affect the values of the parameters corresponding to this material. Slight differences in the biogas yields of the sugar beet pulp were caused by the fact that there were various sugar beet species in this substrate.

### 3.3. pH—Industrial-Scale Measurements

In the PT tank, waste potatoes were comminuted and solid contaminants were removed. They were mixed and liquefied with pig manure (see Section 2.4). During the entire experiment, the pH in the PT tank ranged from 6.91 (3PT) to 7.34 (6PT). Samples were continuously fed from the PT tank to F1 and F2 (primary and secondary digestion), where all the substrates used in the experiment were mixed and underwent primary (F1) and secondary digestion (F2). There were the following pH values in the digesters: ranging from 7.02 (6F1) to 7.23 (1F1) in the first tank and from 7.19 (5F2) to 7.21 (6F2) in the secondary tank, as shown in Figure 1.

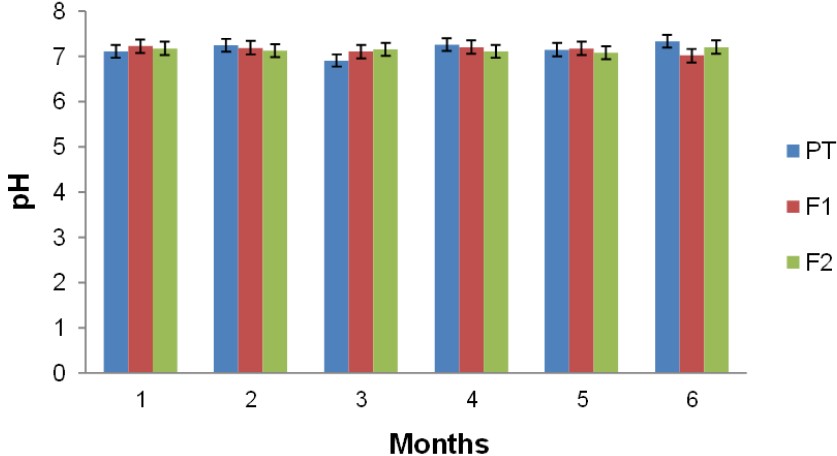

**Figure 1.** Industrial-scale pH measurements in each month of the experiment.

### 3.4. Temperature—Industrial-Scale Measurements

The large volume of liquid manure fed into the predigester significantly reduced the temperature, which was the lowest in this tank. This tank was not intended for the methane fermentation process. The temperature ranged from 12.4 °C to 18.2 °C for preliminary tank. The temperature in the first tank ranged from 38.4 °C to 40.3 °C, whereas in the secondary tank it ranged from 39.4 °C to 40.4 °C.

### 3.5. The Actual Amount of Substrate Fed each Month

The amount of substrates to be fed into the installation was planned on the basis of the results of laboratory tests. The amount was sufficient to generate a power of 1 MW$_{el}$. Depending on the month,

the mass of maize silage fed into the installation ranged from 1668 Mg to 1910 Mg, pig manure—from 4540 Mg to 5160 Mg, waste potatoes—from 62 Mg to 186 Mg, and beet pulp—from 165 Mg to 217 Mg.

### 3.6. The Mass of Methane Produced from Each Substrate Each Month

Methane is one of the most important components of the biogas mixture as it is responsible for its calorific value.

Figure 2 shows the mass of methane obtained from each substrate in each month of the study.

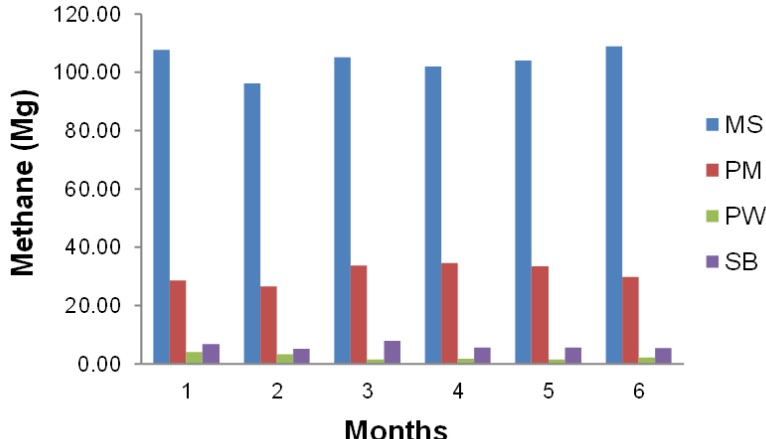

**Figure 2.** The mass of methane produced from the substrates in each month of the experiment.

The installation was equipped with devices that measured the volume of biogas produced and its percentage (volumetric) composition. The biogas volume was converted into mass. The resulting mass of methane refers to each month of the experiment. The average methane content in biogas ranged from 50.4% to 52.1%. These are standard values for substrates used in biogas installations.

### 3.7. Biomass Conversion in Laboratory

The degree of biomass conversion into biogas (methane and carbon dioxide) is important for the efficiency of the process. The content of methane in biogas is important because this compound determines its calorific value. The higher the methane volume concentration is, the less biogas is needed to produce the same amount of energy. It also reduces the demand for substrates because a smaller amount of inert gas (carbon dioxide) is necessary for the combustion process. Therefore, it is important to optimise the process in terms of the methane content in biogas. By testing the biochemical methanogenic potential of a particular organic material, it is possible to select the right amount of substrates for installations capable of generating a specific power.

Figure 3 shows the degree of biomass conversion into biogas (calculations based on Equation (1)). The maize silage was characterised by the best values of its conversion into biogas in the laboratory. However, the distribution of this biomass varied in individual periods of analysis and ranged from 78.8% (2) to 84.2% (6). The highest data amplitude was noted for pig manure, i.e., from 59.8% (6) to 76.9% (4). This indicates that the pig manure used in the test was not homogeneous and contained chemical compounds inhibiting the methane fermentation process, e.g., antibiotics and heavy metals from the feed provided to animals. The lowest degree of biomass conversion into biogas was noted for waste potatoes, i.e., from 43.54% (3) to 45.49% (2). Such poor results indicate that the potatoes may have contained residues from plant protection products, which inhibited biodegradation processes. Moreover, the potatoes were stored in open ground, which caused their decomposition. The degree of the conversion of beet pulp into biogas ranged from 55.90% (4) to 58.86% (2). This percentage of conversion was caused by the composition of organic matter contained in the pulp.

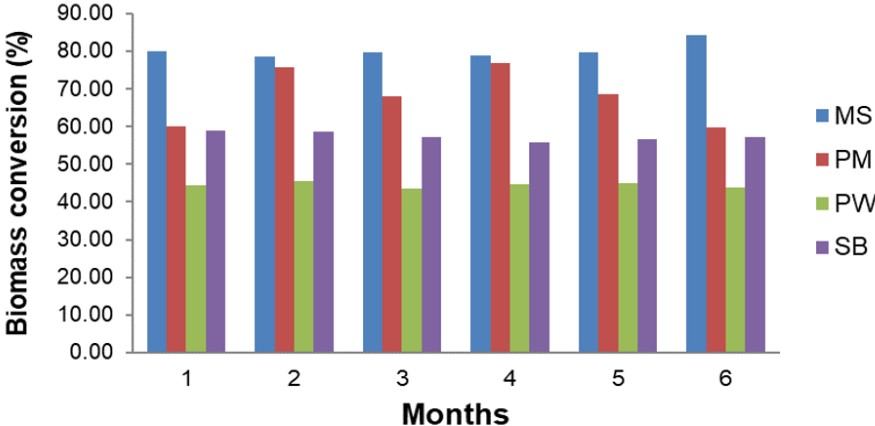

**Figure 3.** Biomass conversion in a laboratory.

*3.8. Biomass Conversion under Industrial Conditions*

In order to verify the degree of biomass conversion under the operating conditions of the installation, the amount of biogas obtained from each substrate was measured. It is easy to calculate the amount of methane from the biogas composition (Figure 4).

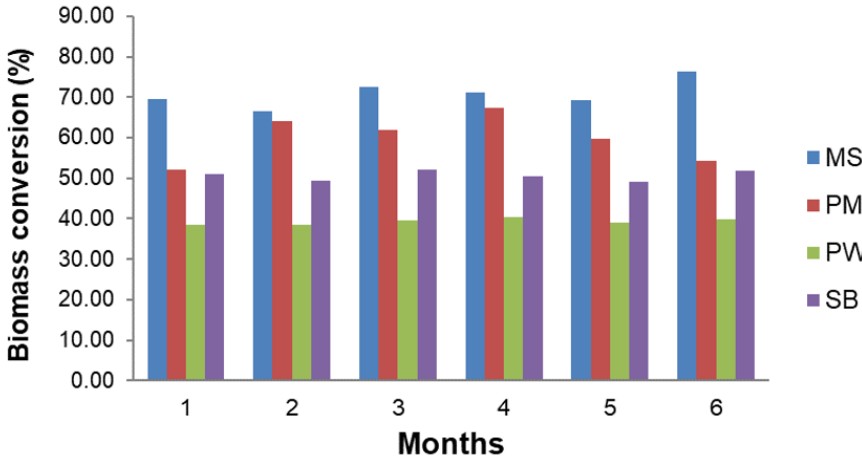

**Figure 4.** Biomass conversion under industrial conditions.

The values of biomass conversion conducted on a technical scale (Equation (2)) were lower than the results noted in the laboratory. This situation was caused by the fact that model tests are conducted in laboratories. However, by analysing the results of biomass conversion on an industrial scale, it is possible to verify the operation of the installation and implement a recovery plan to improve its efficiency. This procedure protects the owners of biogas plants from financial loss because each tonne of biomass which has not been converted into biogas/methane generates additional handling costs. The industrial-scale conversion of maize silage into biogas ranged from 66.44 (2) to 76.46 (6). For pig manure, it ranged from 52.10 (1) to 67.40 (4). Similar to the laboratory tests, the lowest conversion was obtained for waste potatoes, i.e., from 38.41 (2) to 40.27 (4). The conversion of beet pulp into biogas ranged from 49.17 (5) to 51.88 (6). Figure 5 shows the values of biomass conversion into biogas during the operation of the installation.

As resulted from the BMPCC values (Equation (3)), in comparison with the amount of methane obtained under technical conditions, the volume of methane produced in the laboratory was overestimated. There were differences noted for each substrate (see Figure 5). They ranged from 4.7% (6) to 17.19% (2) for maize silage, from 1.14% (4) to 23.58% (1) for pig manure, from 9.5% (4) to 13.69% (2) for waste potatoes, and from 9.06% (3) to 14.31% (5) for beet pulp.

To sum up, the biochemical methanogenic potential of the substrates used in the laboratory investigations was comparable to the data provided in reference publications [51–54]. The values of biogas and methane obtained in the industrial-scale production were lower than in the laboratory tests due to the disturbances that occurred in real installations. However, biogas plant owners should attempt to achieve a comparable efficiency of their facilities to the amounts obtained in laboratory tests. Model tests are used to estimate the amount of batch for the installation and the economic result of the project. Excessive losses of industrial-scale production cause owners to bear additional costs due to the need to purchase additional substrates.

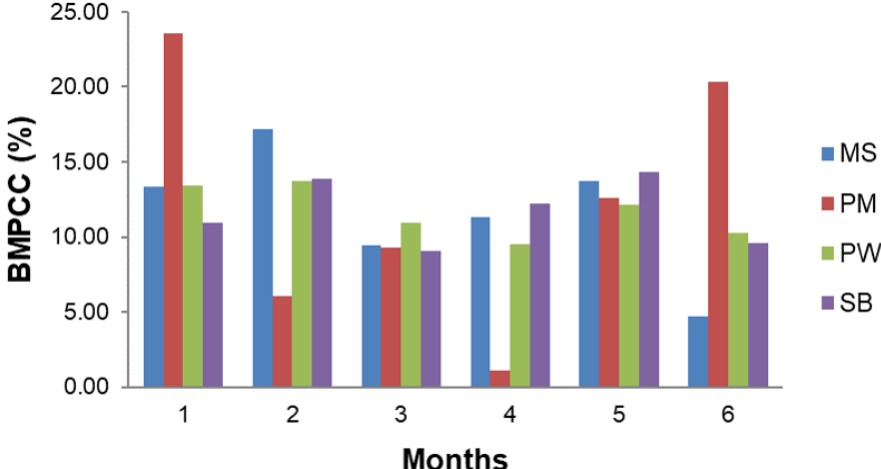

**Figure 5.** Values of the Biochemical Methane Potential Correction Coefficient for the substrates tested.

## 4. Conclusions

The BMPCC enables estimation of the industrial scale under fermentation of biomass compared with the laboratory conditions. Thus, it is possible to analyse the operation of the installation more accurately and eliminate the substrates that may inhibit the methane fermentation process. Waste potatoes seemed to be such a substrate in the installation analysed in this study because they were poorly converted in the laboratory. They may have been contaminated with the substances that weakened the methane fermentation process. When the substrate was fed into the digestion chambers, it may have reduced the process efficiency. In consequence, the biomass conversion in the entire bioreactor was affected, as was proved by the BMPCC.

The Biochemical Methane Potential Correction Coefficient is a solution that can be used as a system diagnosing a methane fermentation plant in a specific (selected) time interval. It can also be used to verify the biochemical methanogenic potential of individual substrates.

**Author Contributions:** Conceptualization, K.P.; methodology, K.P. and A.A.P.; software, P.B. and G.N.; validation, G.N. and P.B.; formal analysis, A.A.P. and K.D.; investigation, K.P., K.W., and N.M.; resources, K.P. and A.A.P.; data curation, K.P., A.A.P., and I.K.; writing—original draft preparation, K.P.; writing—review and editing, A.A.P. and P.B.; visualization, G.N. and P.B.; supervision, K.D.; project administration, K.P. and A.A.P.; funding acquisition, K.P. and K.D. All authors have read and agreed to the published version of the manuscript.

**Funding:** This research was co-funded by the company Biolab-Energy A&P, Poznan', Poland.

**Conflicts of Interest:** The authors declare no conflict of interest.

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
