# Peer review of "The Efficiency of Industrial and Laboratory Anaerobic Digesters of Organic Substrates: The Use of the Biochemical Methane Potential Correction Coefficient"

_energies, doi:10.3390/en13051280_

Round 1
Reviewer 1 Report
In the present manuscript, written by Pilarski et al, authors present a methodology to determine the Biochemical Metane Potential Correlation Coefficient based on the calculation of biomass conversion measured in two different working scales. The development of correlations that allow to predict the behavior of scale-up systems is highly desirable but still challenging, due to the diversity of parameters involved. In this context, the topic is relevant, and although the study is clearly within the scope of journal, I’m not fully convinced about the necessary scientific novelty for publication. I believe it needs several corrections to be more acceptable for publication. In my opinion authors should rethink the follow:
Comments:
1. Title should be more focus on the work. In my opinion is too generic.
2. The abstract should be rewritten. The authors are honest in informing that some data presented is part of a work already presented, however the focus should be on the novelty of study. This information may be given in another way. The “meaning” of technical scale should be clear. Please do not put references in the abstract.
3. Authors should avoid, in all the manuscript, expressions like: “The authors of this study”; “results of our experiment” and “authors of reference publications”, please choose impersonal expressions.
4. Pag 3 - Line 102: “The following (…) substrates and the samples”. Please be clear about the difference of substrates and samples, if there is any difference. It´s possible they are samples of the substrates?
5. Section 3.1. and 3.2:
- Authors should check the values of Table 1 whit the values presented in the text (minima and maxima) and specially the VS content of PW. Also, in the Table the samples of SB should be correct from 1 SB to the respective 1 to 6.
- If the samples, for each month, result from the analysis of three samples, would it not be possible to place the standard deviation values in Table 1?
- I would also like to see a discussion of the results, instead of a transcription of the results presented in Table 1. Authors use other studies to make valuable comparations, however the influence of the parameters in the process is not discussed.
6. In section 3.3. please be clear about the substrates.
7. Could a schematic representation of the industrial plant be included? Perhaps would help to understand the dynamics and visualize the process.
8. Page 1 - Line 36: The The, please erase.
Author Response
Dear Reviewer 1
Thank you for your valuable comments.
- Title should be more focus on the work. In my opinion is too generic.
- The title has been changed so as to provide more details:
‘The Efficiency of Industrial and Laboratory Anaerobic Digesters of Organic Substrates: The Use of the Biochemical Methane Potential Correction Coefficient’
- The abstract should be rewritten. The authors are honest in informing that some data presented is part of a work already presented, however the focus should be on the novelty of study. This information may be given in another way. The “meaning” of technical scale should be clear. Please do not put references in the abstract.
- As was agreed and recommended by the editor, the article was supposed to be a CONTINUATION of the conference paper cited as [1]. The editor recommended that the conference participants and the authors of proceedings should mention this in the first sentences of the article. I have removed this citation from the Abstract, but I must stress that the Abstract describes the article only.
- Authors should avoid, in all the manuscript, expressions like: “The authors of this study”; -
- As recommended, all these expressions have been removed.
- Pag 3 - Line 102: “The following (…) substrates and the samples”. Please be clear about the difference of substrates and samples, if there is any difference. It´s possible they are samples of the substrates?
- Substrates were pure materials used in the experiment, whereas samples were pieces of the fermenting mixture that were collected for analyses during the operation of the installation.
- Section 3.1. and 3.2:
- Authors should check the values of Table 1 whit the values presented in the text (minima and maxima) and specially the VS content of PW. Also, in the Table the samples of SB should be correct from 1 SB to the respective 1 to 6.
- All the values have been checked. The wrong values copied from the Table have been corrected. Thank you for your remark.
- If the samples, for each month, result from the analysis of three samples, would it not be possible to place the standard deviation values in Table 1?
- The values in Table 1 have been marked as uncertain.
- I would also like to see a discussion of the results, instead of a transcription of the results presented in Table 1. Authors use other studies to make valuable comparations, however the influence of the parameters in the process is not discussed.
- As recommended, the results provided in Table 1 have been discussed briefly (lines 314-323).
The pH range corresponding to the maize silage proved that the ensilage process was successful. During storage the total solids as well as volatile solids of this substrate were reduced. It is most likely that it was caused by microorganisms responsible for the conservation of MS. However, despite the decrease in the MS organic matter, the amount of biogas produced in the following months was not reduced. The physicochemical properties of pig manure depend on the way animals are fed and their age. This significantly affects the biomethane efficiency of this material. It is similar with potato waste, which is a waste material but it is not cultivated for energy purposes. However, as the table shows, this did not affect the values of the parameters ​​corresponding to this material. Slight differences in the biogas yields of the sugar beet pulp were caused by the fact that there were various sugar beet species in this substrate.
- In section 3.3. please be clear about the substrates.
In the PT tank waste potatoes were comminuted and solid contaminants were removed. They were mixed and liquefied with pig manure (see 2.4.). During the entire experiment the pH in the PT tank ranged from 6.91 (3PT) to 7.34 (6PT). Samples were continuously fed from the PT tank to F1 and F2 (primary and secondary digestion), where all the substrates used in the experiment were mixed and they underwent primary (F1) and secondary digestion (F2). There were the following pH values in the digesters:…………………………….(lines 327-333).
- Could a schematic representation of the industrial plant be included? Perhaps would help to understand the dynamics and visualize the process.
- There is a short description of the installation in section. 2.4. We regret to say that due to the confidential nature of data (agreements between the owner of the biogas plant and the University) we cannot provide a photo or a diagram of the installation.
- Page 1 - Line 36: The The, please erase.
- We have erased ‘The’.
Best regards,
Agnieszka Pilarska

Reviewer 2 Report
The authors described the methodology of defining the Biochemical Methane Potential Correction Coefficient (BMPCC) based on the calculation of biomass conversion on an industrial scale and on a laboratory scale. This study compares the efficiency of the anaerobic digestion technology applied for maize silage (MS), pig manure (PM), potato waste (PW) and sugar beet pulp (SB) on an industrial scale with its efficiency on a laboratory scale. The comparison was made for the same substrates. They also also verified the level of significance of the determinants affecting the efficiency of operation of the biogas plant. I have a few comments to improve further the quality of the manuscript.
Introduction:
Include the information of disturbances that occurred in real installation and how they can effect on biogas numbers.
Include few sentence for biogas process and microbial community when maize silage (MS), pig manure (PM), potato waste (PW) and sugar beet pulp (SB) were used. Cite papers for effect of pH and temperature on methane production and microbial community. It would be interesting to see what microbial community is positively affecting on methane production.
Line 36 remove “The”
2.1 Materials:
“The waste materials were acquired from a farm and a sugar factory in the Wielkopolska region, Poland.” Include the month and location. What is the process for sample collection?
Include the maize silage (MS), pig manure (PM), potato waste (PW) and sugar beet pulp (SB) storage conditions.
2.4 What are the factors that effects on physicochemical properties after storage?
Line 146- Remove repetitive word Biochemical Methane Potential Correction Coefficient (BMPCC).
Line 189- What’s the unit of the concentration of hydrogen ions?
Table 1, include the statistical errors in the table. How many technical or biological replicates performed?
Section 3.3 and 3.4:
Describe how pH and Temperature effect on methane production and microbial community.
Line 286: What’s the composition of biogas?
Line 342: What are the disturbances that occurred in real installation? Explain it in brief.
Author Response
Dear Reviewer 2
Thank you for your valuable comments.
Introduction:
- Include the information of disturbances that occurred in real installation and how they can effect on biogas numbers.
In the Introduction we included information about the disturbances on the installation which may have significantly affected the amount of biogas/methane produced – it was lower than the amount produced on the laboratory scale. We have supplemented the information as suggested.
A slight change in the conditions of the methane digestion in a biogas plant may upset the process or halt it completely. Temperature is one of the factors that significantly affects the course of the digestion process. Small changes in temperature and the retention time affect bacterial activity [13]. In consequence, the biogas production is reduced. In Poland the digestion process is usually carried out under mesophilic conditions at a temperature of about 39°C. In Europe mesophilic installations are predominant, but there are also thermophilic conditions, where more heat energy is used by the plant to sustain the process [14, 15]. As was mentioned before, pH is also important for methane fermentation. It should range from 6.8 to 7.5, because this guarantees optimal conditions for bacterial life and reproduction during all the four phases of the methane fermentation process [16, 17]. The biogas yield decreases when the pH value is higher than 7.5 or lower than 6.8.
When biomass is applied to digestion chambers with high-protein substrates, excessive amounts of ammoniacal nitrogen are usually formed. Excess ammonia significantly slows down the biogas production process. As temperature increases, so does the effect of ammonia inhibiting methane fermentation [18, 19]. Some substrates are rich in sulphur and they cause excessive amounts of this element in the digester. The element may occur in the form of ions dissolved during the liquid phase or it may be found in hydrogen sulphide in a mixture of liquid and gas. Higher temperature increases the solubility of hydrogen sulphide, which results in its higher concentration during the liquid phase [20]. At the initial stage of operation of a biogas plant it is very important to supply small portions of substrates with a homogeneous composition. This affects the overall normal digestion by individual bacteria at each stage of the fermentation process [21]. During the operation of a biogas plant it is necessary to observe the retention time for individual organic substrates. It is the time a given substrate should spend in the digester until it achieves an appropriate level of degradation. These are the most common problems encountered in practice, which directly affect the actual production of biogas.
Line 36 remove “The”
We have erased ‘The’.
2.1 Materials:
- “The waste materials were acquired from a farm and a sugar factory in the Wielkopolska region, Poland.” Include the month and location. What is the process for sample collection?
We regret to say that these data are considered sensitive. Access to them has been restricted by the owner of the biogas plant where the experiment was conducted.
- Include the maize silage (MS), pig manure (PM), potato waste (PW) and sugar beet pulp (SB) storage conditions.
Maize silage was stored in silos, where it was ensilaged. Pig manure was supplied directly from a pig farm. Potato waste was also supplied directly after being customised. Sugar beet pulp was stored in silage sleeves.
- 2.4 What are the factors that effects on physicochemical properties after storage?
The degree of maize compaction in the ensilage process, the weather conditions and storage time were the factors affecting the physicochemical properties after storage.
- Line 146- Remove repetitive word Biochemical Methane Potential Correction Coefficient (BMPCC).
We have removed the repetitive word.
- Line 189- What’s the unit of the concentration of hydrogen ions?
As a chemist, I don't understand the question. The pH is defined as the concentration of hydrogen ions and it has no unit.
- Table 1, include the statistical errors in the table. How many technical or biological replicates performed?
I this study, she safest and most practical station for manual sample collection was selected. The practicality of this location was analysed in terms of the representativeness of the material collected for tests. Each time at least 3 samples were collected to increase confidence in the representativeness of the material collected.
The values of uncertainty of the results have been supplemented in Table 1.
Uncertainty is a basic property of each measurement. It is found at every stage of the measurement procedure. Usually, when using an appropriate analytical procedure to test samples, uncertainty may be caused by [36]: incorrect or imprecisely defined size of measurement, unrepresentativeness of the sample collected, incorrect measurement method, uncertainty caused by calibration of the measuring device, uncertainty related with the measurement standards and/or reference materials, fluctuations in measurement replicates despite seemingly identical external conditions.
It is necessary to clarify the difference between a measurement error and uncertainty. An error is the difference between the value measured and the expected value. Uncertainty is the range within which the expected value is likely to be found [37]. When the uncertainty of measurement is determined, its reliability increases. In consequence, it is possible to compare measurements between laboratories.
When estimating the uncertainty of measurement in this article, the procedures followed Polish standards and one German standard [35].
- Section 3.3 and 3.4:
Describe how pH and Temperature effect on methane production and microbial community.
The information has been provided in the Introduction and briefly mentioned in Table 1.
- Line 286: What’s the composition of biogas?
The average methane content in biogas ranged from 50.4% to 52.1%. These are standard values for substrates used in biogas installations.
- Line 342: What are the disturbances that occurred in real installation? Explain it in brief.
The information has been provided in the Introduction, as Reviewer #1 also requested.
Best regards,
Agnieszka Pilarska

Round 2
Reviewer 1 Report
Dear authors
Thank you for your response. I read the second version and I consider that all my questions were addressed. I only have the two following small recommendations prior publication:
1) In my opinion, the definitions presented in Section “2.8. Uncertainty of the results” are not necessary!
2) Results, in Table 1, should appear for all parameters as: value±UR (4.21±0.06) and not with UR as a new column. Please remake the Table.
Author Response
Dear Reviewer 1,
- I removed section 2.8. ‘Uncertainty of the results’.
- I remaked the Table.
Thank very much you.
Best regards,
Agnieszka Pilarska
